# The impacts of parity on lung function data (LFD) of healthy females aged 40 years and more issued from an upper middle income country (Algeria): A comparative study

**Abdelbassat Ketfi** [1], **Leila Triki** [2], **Merzak Gharnaout** [1], **Helmi Ben Saad** [3,4,5] *

**1** Department of Pneumology, Phtisiology and Allergology, Rouiba Hospital, Algiers, University of Algiers 1, Faculty of Medicine, Algiers, Algeria, **2** Department of Functional Explorations, Habib-Bourguiba Hospital, University of Sfax, Faculty of Medicine of Sfax, Sfax, Tunisia, **3** Department of Physiology and Functional Explorations, University of Sousse, Farhat Hached Hospital, Sousse, Tunisia, **4** Laboratory of Physiology, Faculty of Medicine of Sousse, University of Sousse, Sousse, Tunisia, **5** Heart Failure Research Laboratory (LR12SP09), Farhat Hached Hospital, University of Sousse, Sousse, Tunisia

* helmi.bensaad@rns.tn

**Data Availability Statement:** All relevant data are within the paper and the Supporting Information file 4 (S4 File).

## Abstract

### Background

Studies evaluating the impacts of parity on LFD of healthy females presented controversial conclusions.

### Aim

To compare the LFD of healthy females broken down according to their parities.

### Methods

A medical questionnaire was administered and anthropometric data were determined. Two groups [$G_1$ (n = 34): ≤ 6; $G_2$ (n = 32): > 6] and three classes [$C_1$ (n = 15): 1–4; $C_2$ (n = 28): 5–8; $C_3$ (n = 23): 9–14] of parities were identified. LFD (plethysmography, specific airway resistance (sRaw)] were determined. Student's t-test and ANOVA test with post-Hoc test were used to compare the two groups' and the three classes' data.

### Results

$G_1$ and $G_2$ were age and height matched; however, compared to $G_1$, $G_2$ had a lower body mass index (BMI). $C_1$, $C_2$ and $C_3$ were height, weight and BMI matched; however, compared to $C_2$, $C_3$ was older. $G_1$ and $G_2$ had similar values of $FEV_1$, forced- and slow- vital capacities (FVC, SVC), maximal mid-expiratory flow (MMEF), forced expiratory flow at x% of FVC ($FEF_{x\%}$), peak expiratory flow (PEF), expiratory and inspiratory reserve volumes (ERV, IRV, respectively), inspiratory capacity (IC), sRaw, $FEV_1$/FVC, $FEV_1$/SVC, and residual volume/total lung capacity (RV/TLC). The three classes had similar values of MMEF, $FEF_{x\%}$, PEF, thoracic gas volume (TGV), ERV, IRV, $FEV_1$/FVC, $FEV_1$/SVC and RV/TLC. Compared to $G_1$, $G_2$ had higher TGV (2.68±0.43 vs. 3.00±0.47 L), RV (1.80±0.29 vs. 2.04±0.33 L) and

**Funding:** The author(s) received no specific funding for this work.

**Competing interests:** Helmi BEN SAAD reports personal fees from AstraZeneca, Boehringer Ingelheim, INPHA-MEDIS, Teriak, Chiesi, SAIPH and HIKMA. This does not alter our adherence to PLOS ONE policies on sharing data and materials. The remaining authors declare that they have no conflicts of interest concerning this article.

**Abbreviations: ATS**, American thoracic society; **BMI**, body mass index; **C**, class; **COPD**, chronic obstructive pulmonary disease; **ERS**, European respiratory society; **ERV**, expiratory reserve volume; **FEF$_{x\%}$**, forced expiratory flow when x% of FVC remains to be exhaled; **FEV$_1$**, forced expiratory volume in 1 s; **FVC**, forced vital capacity; **G**, group; **GLI**, global lung initiative; **IC**, inspiratory capacity; **IRV**, inspiratory reserve volume; **LFD**, lung function data; **MMEF**, maximal mid-expiratory flow; **OVD**, obstructive ventilatory defect; **PEF**, peak expiratory flow; **r²**, determination-coefficient; **RV**, residual volume; **SD**, standard deviation; **sRaw**, specific airway resistance; **SVC**, slow vital capacity; **TGV**, thoracic gas volume; **TLC**, total lung capacity.

TLC (4.77±0.62 vs. 5.11±0.67 L). Compared to $C_1$, $C_2$ had higher FEV$_1$ (2.14±0.56 vs. 2.47±0.33 L), FVC (2.72±0.65 vs. 3.19±0.41 L), SVC (2.74±0.61 vs. 3.24±0.41 L), TLC (4.47±0.59 vs. 5.10±0.58 L), IC (1.92±0.41 vs. 2.34±0.39 L) and sRaw (4.70±1.32 vs. 5.75±1.18 kPa*s). Compared to $C_1$, $C_3$ had higher TLC (4.47±0.59 vs. 5.05±0.68 L) and RV (1.75±0.29 vs. 2.04±0.30 L).

## Conclusion

Increasing parity induced a tendency towards lung-hyperinflation.

## Introduction

Respiratory aging, which would start from the age of 35–40 [1], can be estimated from lung function data (LFD), recognized as predictors of mortality and morbidity [2]. Indeed, the American thoracic and the European respiratory societies (ATS/ERS) recommended the use of LFD to diagnose any respiratory defect, even before any clinical manifestation [3]. In a healthy and asymptomatic population, the decline in LFD is associated with a high risk of cardiopulmonary diseases and all other causes of death [4]. Since the different aspects of LFD' decline are still inescapable [5], and since the determination of numerous factors of LFD' decline is ongoing, their analysis is necessary to formulate strategies to prevent lung-aging, especially in females without clinical symptoms [6].

In practice, once lung function test quality has been validated, the succeeding step consists in comparisons of measured/determined LFD with data generated from reference equations based on healthy subjects [3]. On the one hand, these equations are based on the subject's ethnicity and included as influencing factors sex and some anthropometric data [*eg*, age, height, weight and body mass index (BMI)] [3, 7, 8]. On the other hand, LFD' influencing factors are numerous and are not limited to the aforementioned characteristics, which explain only 70% of their variance [9]. According to the literature, the influence of different characteristics/factors on the variance of the forced vital capacity (FVC) is ± 30% for sex, 20% for height, 10% for group ethnic, 8% for age, 3% for technical factors, 2% for weight, and the remaining 30% for other factors [*eg*, air pollution, climatic conditions, altitude, socioeconomic-level, schooling-level, physical-activity level, thoracic diameter, nutritional status] [9, 10]. Among the remaining LFD influencing factors, parity has been proposed in some studies [6, 7, 11–22]. On the one hand, contrary to high income countries such as European and North American ones, parity is a particular issue in low- and lower-middle-income countries such as African ones [23]. For example, during 2015, while the European and the Asian parity means were respectively, 1.616 and 2.173, that of Africa was 4.589 [23] (S1 File). Moreover, in some African countries, the 2015 mean values of parity were higher than six [*eg*, 6.365, 6.202, 6.145 and 6.050, respectively, in Chad, Somalia, Democratic Republic of Congo and Mali] [23] (S1 File). To the best of the authors' knowledge, few studies, published between 1999 and 2018, have raised the impact of parity on healthy females' LFD [USA (n = 1 [11]), Tunisia (n = 3 [6, 7, 12]), Nigeria (n = 2 [14, 16]), Brazil (n = 1 [15]] with contradictory results. On the one hand, some studies concluded that high parity was associated with positive effects on LFD of American [*eg*, larger forced expiratory volume in 1 s (FEV$_1$) and FVC [11]] or of Nigerian [*eg*, increases in FEV$_1$ and FVC [14], even across all females positions [16]] In the other hand, some other studies concluded that high parity was associated with negative effects on LFD of Tunisian [*eg*, reduction in peak flow rate [7], a tendency towards a proximal obstructive ventilatory defect (OVD)

[6] and acceleration of lung-aging [12, 13]], or of Brazilian aged < 25 years [**eg**, lower maximal mid-expiratory flow (MMEF) and peak expiratory flow (PEF) [15]]. Moreover, a North-African study concluded that compared to aging by one year, the parity increase of one unit, caused a greater LFD' decline [**eg**, $FEV_1$ declines were 23 and 33 mL, respectively, per year of age and when parity increases by one unit [6]].

In view of the above divergence between studies, the main aim of the present study was to compare the LFD of healthy North-African females broken down according to their parities into two groups [$G_1$: parity $\leq 6$; $G_2$: parity $> 6$] and three classes [$C_1$: parity $\leq 4$; $C_2$: $5 \leq$ parity $\leq 8$; $C_3$: parity $\geq 9$]. The second aim was to determine the relationship between parity and some LFD.

## Population and methods

This present study is part of a project involving four parts. The first, which was recently published [24], aimed at testing the applicability of the global lung initiative (GLI-2012) norms on a sample of healthy adults living in Algiers. The second, recently published [10], aimed to test the applicability of the Eastern Algeria plethysmographic norms [8] on a sample of healthy adults living in Algiers. The third part is the objective of this study. The fourth part, will be the establishment, according to recent international recommendations [25], of plethysmographic norms specific to the population of northern Algeria. For the above reasons, a large part of the methodology of this study has already been the object of previous descriptions [10, 24].

### Study design

It was a comparative study performed in the Department of Pneumology, Phtisiology and Allergology at the Rouiba Hospital, Algiers. The study was conducted in compliance with the 'Ethical principles for medical research involving Human subjects of the Helsinki Declaration (https://www.wma.net/wp-content/uploads/2016/11/ethics_manual_arabic.pdf; last visit: September 25[th] 2019). The study was approved by the Rouiba Hospital (Algiers) Medical Council and Ethics Committee (approval number: 0601/2014). Written informed consent was obtained from all participants who were not charged any costs for the accomplished tests.

### Sample size

The null hypothesis [26] was H0: $m_1 = m_2$, and the alternative one was Ha: $m_1 = m_2 + d$, where "d" is the difference between two means and $n_1$ and $n_2$ are the sample sizes for the two groups ($G_1$ and $G_2$) of females, such $N = n_1 + n_2$. The sample size was estimated using the following formula [26]:

$$N = [(r + 1)(Z_{\alpha/2} + Z_{1-\beta})^2 \delta^2]/(R\,d^2)$$

- "$Z_{\alpha/2}$" is the normal deviate at a level of significance = 1.64 (0.10 level of significance);

- "$Z_{1-\beta}$" is the normal deviate at 1-β% power with β% of type II error (0.84 at 80% statistical power);

- "R" (= $n_1/n_2$) is the ratio of sample size required for two groups (R = 1 gives the sample size distribution as 1:1 for two groups);

- "s" and "d" are the pooled standard-deviation (SD) and difference of total lung capacity (TLC) means of two groups. These two values were obtained from a Tunisian study

including females aged $\geq$ 60 years [6] where TLC means of two groups of females (parity < 4 vs. parity $\geq$ 4) were, respectively, 5.01 and 4.62 L, with a common SD equal to 0.92 L. The sample size for the study was 68 (34 females in each group).

## Population: Inclusion, non-inclusion and exclusion criteria

The target population consisted of a group of healthy adults aged $\geq$ 18 years. These adults were selected by convenience sampling among visitors and the acquaintances of hospitalized patients in the aforementioned Department. The population was relatively homogeneous and considered as belonging to the middle class with an elevated human development index of 0.75 [27]. The latter is a measure of the average quality of life of a country's population, ranging from 0 to 1, and takes into account three dimensions of human development (life expectancy, years of schooling, and income).

Only healthy females aged $\geq$ 40 years with at least one parity and presenting technically acceptable and reproducible plethysmographic maneuvers were included in this study. The following non-inclusion criteria were applied: *(i)* acute or chronic diseases of the respiratory system [*eg*, asthma, chronic bronchitis, chronic obstructive pulmonary disease (COPD), emphysema, tuberculosis] or previous hospitalization for pulmonary or thoracic problems; *(ii)* cardiac diseases that may affect the respiratory system [*eg*, heart failure, arrhythmia, unstable angina or myocardial infarction, uncontrolled high blood pressure]; *(iii)* current-smoker or ex-smoker of more than one pack-year; *(iv)* leanness and obesity stage 2 and more; and *(v)* high physical-activity level [*eg*, sports practice > 5 h/week [6]].

The total population was divided into four groups: a group for the development of plethysmographic norms specific to northern Algeria' population (n = 491, 49.7% females), a group for the validation of the GLI-2012 spirometric norms (n = 300, 50.0% females) [24], a group for the validation of the Eastern Algeria plethysmographic norms (n = 453, 51.7% females) [8], and this study group (n = 66 females).

## Collected data

Clinical data were collected using the ATS questionnaire [28], widely described elsewhere [8, 10, 24]. Parity, defined as the number of offspring a female has borne, was introduced in three forms: numerical (unit), two groups [$G_1$ (parity $\leq$ 6); $G_2$ (parity > 6)] and three classes [$C_1$: 1 $\leq$ parity $\leq$ 4; $C_2$: 5 $\leq$ parity $\leq$ 8; $C_3$: parity $\geq$ 9] [17, 29–31]. Menopausal status was determined using the stages of reproductive aging workshop classification [32]. Females were classified into premenopausal (regular or irregular menses) or postmenopausal (lack of menses for over one year or hysterectomy).

The decimal age was calculated from the date of measurement and the date of birth. Standing height (m) and weight (kg) were recorded and BMI (kg/m$^2$) was calculated. Obesity status was categorized into leanness (BMI < 18.5), normal weight (18.5 $\leq$ BMI $\leq$ 24.9), overweight (25.0 $\leq$ BMI $\leq$ 29.9), and obesity stages 1 (30.0 $\leq$ BMI $\leq$ 34.9), 2 (35.0 $\leq$ BMI $\leq$ 39.9) and 3 (BMI > 40.0) [33].

## Plethysmographic measurements

LFD were determined by one qualified person (*AK in the authors' list*) via a plethysmograph (Body-box 5500, MediSoft, Belgium). The latter was calibrated each morning. The following data were determined: flow-volume curve' data [FVC (L), FEV$_1$ (L), PEF (L/s), MMEF (L/s), forced expiratory flow when x% FVC remains to be exhaled (FEF$_{x\%}$, L/s)], volumes and

capacities [expiratory reserve volume (ERV, L), inspiratory reserve volume (IRV, L), inspiratory capacity (IC, L), slow vital capacity (SVC, L), residual volume (RV, L), TLC (L), thoracic gas volume (TGV, L)], ratios (FEV$_1$/FVC, FEV$_1$/SVC, RV/TLC, absolute values), and specific airway resistance (sRaw, kPa$^*$s).

The plethysmographic measurements were performed according to the international recommendations [34, 35], widely described elsewhere [8, 10, 24, 36], and the reproducibility and acceptability criteria were respected [34, 35]. The acceptability and reproducibility of the FVC maneuvers were described elsewhere [24]. Regarding the TGV repeatability, at least three values were obtained so that the difference between the highest and the lowest TGV values divided by the mean was ≤ 0.05 [35]. The TGV average value was selected [35].

## Statistical analysis

Qualitative and quantitative data were expressed by their mean±SD and relative frequencies. Student's t-test and Chi-square test were used to compare, respectively, the two groups' of parity quantitative and qualitative data. An analysis of variance with post-Hoc test was carried out for the three parity classes' quantitative data. The Pearson Chi-square test was used to compare the obesity and menopause statuses of the three parity classes. When applicable, significant differences between percentages were tested using the McNemar test. Determination-coefficient (r$^2$ = square of the Pearson product-moment correlation-coefficient) evaluated the associations between parity, and plethysmographic and anthropometric data. "r$^2$" was considered as "clinically significant" when it was > 0.30 [37]. Hedge's TLC value was used for effect size measurement [38]. An effect size of ≤ 0.2 was described as a small effect, around 0.5 as medium effect, around 0.8 as a large effect, and more than 1.30 as very large effect [38]. All mathematical computations and statistical procedures were performed using Statistica software (Statistica Kernel version 6; Stat Software. France). Significance was set at 0.05 level.

## Results

Among the 244 females included in the whole project, 121 were excluded from this study (97 were under 40 years of age and 24 were nulliparous). The parity' mean±SD of the remaining 123 females aged ≥ 40 with at least one parity was 6.4±3.5. Based on this data, two groups were randomly chosen to be age- and height- matched [G$_1$ (parity ≤ 6) (n = 34), G$_2$ (parity > 6) (n = 32)]. Fig 1 presents the mean age of the 66 females divided according to their parities.

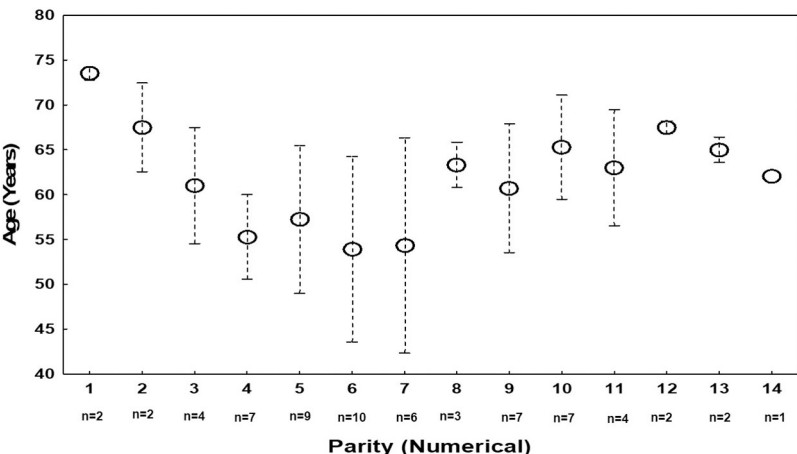

**Fig 1. Mean age of the 66 females broken down according to their parities (numerical data).**

**Table 1. Characteristics and plethysmographic data of the total sample and of females divided into two groups (G) of parities: $G_1$ (parity $\leq$ 6) and $G_2$ (parity > 6).**

| | | Total sample (n = 66) | $G_1$ (n = 34) | $G_2$ (n = 32) | p |
|---|---|---|---|---|---|
| **Parity, anthropometric data, obesity and menopause status** | | | | | |
| **Parity** | numerical | 7±3 | 4±1 | 10±2 | 0.00001 |
| **Age** | Years | 60±9 | 58±9 | 62±8 | 0.0649 |
| **Height** | m | 1.56±0.05 | 1.55±0.06 | 1.56±0.05 | 0.6530 |
| **Weight** | kg | 70±10 | 73±10 | 67±9 | 0.0271* |
| **BMI** | kg/m$^2$ | 28.8±3.7 | 30.0±3.60 | 27.6±3.4 | 0.0073* |
| **Obesity status** | Normal weight | 12 (18.2) | 4 (11.8) | 8 (25.0) | 0.1648 |
| | Overweight | 27 (40.9) | 11 (32.3) | 16 (50.0) | 0.1438 |
| | Obesity stage 1 | 27 (40.9) | 19 (55.9) | 8 (25.0) | 0.0107** |
| **Menopause** | Yes | 58 (87.9) | 28 (82.3) | 30 (93.8) | 0.1526 |
| **Flow-volume curve' data** | | | | | |
| **FEV$_1$** | L | 2.33±0.42 | 2.32±0.49 | 2.33±0.34 | 0.9331 |
| **FVC** | L | 3.01±0.53 | 2.97±0.59 | 3.04±0.47 | 0.6001 |
| **MMEF** | L/s | 3.03±0.61 | 3.11±0.71 | 2.95±0.46 | 0.2925 |
| **FEF$_{25\%}$** | L/s | 0.89±0.44 | 0.92±0.41 | 0.86±0.46 | 0.6167 |
| **FEF$_{50\%}$** | L/s | 3.22±0.90 | 3.31±0.95 | 3.12±0.85 | 0.3987 |
| **FEF$_{75\%}$** | L/s | 5.18±1.02 | 5.30±1.16 | 5.05±0.86 | 0.3089 |
| **PEF** | L/s | 5.72±1.16 | 5.87±1.38 | 5.56±0.87 | 0.2796 |
| **Volumes and capacities** | | | | | |
| **SVC** | L | 3.05±0.53 | 3.01±0.57 | 3.09±0.49 | 0.5457 |
| **TGV** | L | 2.83±0.47 | 2.68±0.43 | 3.00±0.47 | 0.0053* |
| **RV** | L | 1.91±0.33 | 1.80±0.29 | 2.04±0.33 | 0.0026* |
| **TLC** | L | 4.94±0.66 | 4.77±0.62 | 5.11±0.67 | 0.0349* |
| **ERV** | L | 0.88±0.37 | 0.82±0.35 | 0.93±0.38 | 0.2351 |
| **IRV** | L | 1.55±0.42 | 1.60±0.46 | 1.50±0.37 | 0.3776 |
| **IC** | L | 2.17±0.43 | 2.19±0.48 | 2.16±0.38 | 0.8005 |
| **Airway resistances** | | | | | |
| **sRaw** | kPa*s | 5.47±1.26 | 5.34±1.343 | 5.60±1.17 | 0.3906 |
| **Ratios** | | | | | |
| **FEV$_1$/FVC** | absolute value | 0.77±0.04 | 0.78±0.04 | 0.77±0.03 | 0.1801 |
| **FEV$_1$/SVC** | absolute value | 0.76±0.05 | 0.77±0.05 | 0.76±0.05 | 0.2686 |
| **RV/TLC** | absolute value | 0.39±0.05 | 0.38±0.06 | 0.40±0.04 | 0.1743 |

**BMI**: body mass index. **ERV**: expiratory reserve volume. **FEF$_{x\%}$**: forced expiratory volume at x% of FVC. **FEV$_1$**: forced expiratory volume in 1 s. **FVC**: forced vital capacity. **IC**: inspiratory capacity. **IRV**: inspiratory reserve volume. **MMEF**: maximal mid-expiratory flow. **PEF**: peak expiratory flow. **RV**: residual volume. **sRaw**: specific resistance airway. **SVC**: slow vital capacity. **TGV**: thoracic gas volume. **TLC**: total lung capacity. Quantitative data were mean±SD. Qualitative data (obesity and menopause status) were number (%).

*p (student test) < 0.05: $G_1$ vs. $G_2$.

**p (Chi-2 test) < 0.05: $G_1$ vs. $G_2$.

Females with parity equaling 6 (n = 10), 5 (n = 9), 4, 9 and 10 (n = 7 each) dominated the distribution. All together their represented 60.6% of the total sample.

Table 1 exposes the data of the females divided into two groups of parities. The two groups were age- and height- matched, and included similar percentages of menopaused females. Compared to the $G_1$, the $G_2$ was lighter, had a significantly lower BMI, and included a lower percentage of females with obesity stage 1. The two groups had similar values of the flow-

**Table 2. Characteristics and plethysmographic data of the females divided into three classes (C) of parities: $C_1$ (parity $\leq$ 4); $C_2$ (5 $\leq$ parity $\leq$ 8) and $C_3$ (9 $\leq$ parity $\leq$ 14).**

| | | $C_1$ (n = 15) | $C_2$ (n = 28) | $C_3$ (n = 23) | p |
|---|---|---|---|---|---|
| **Parity, anthropometric data, obesity and menopause status** | | | | | |
| **Parity** | numerical | 3±1 | 6±1 | 10±1 | 0.0000*[**abc**] |
| **Age** | Years | 61±8 | 56±10 | 64±6 | 0.0062*[**c**] |
| **Height** | m | 1.56±0.06 | 1.56±0.05 | 1.55±0.05 | 0.7454 |
| **Weight** | kg | 71±10 | 72±10 | 67±9 | 0.0961 |
| **BMI** | kg/m$^2$ | 29.4±4.2 | 29.6±3.7 | 27.6±3.2 | 0.1264 |
| **Obesity status** | Normal weight | 3 (20.0) | 4 (14.3) | 5 (21.7) | 0.2054 |
| | Overweight | 4 (26.7) | 10 (35.7) | 13 (56.4) | |
| | Obesity stage 1 | 8 (53.3) | 14 (50.0) | 5 (21.7) | |
| **Menopause** | Yes | 15 (100.0) | 20 (71.4) | 23 (100.0) | 0.0021**[**ac**] |
| **Flow-volume curve' data** | | | | | |
| **FEV$_1$** | L | 2.14±0.56 | 2.47±0.33 | 2.27±0.37 | 0.0301*[**a**] |
| **FVC** | L | 2.72±0.65 | 3.19±0.41 | 2.96±0.50 | 0.0167*[**a**] |
| **MMEF** | L/s | 2.93±0.75 | 3.17±0.65 | 2.93±0.41 | 0.2788 |
| **FEF$_{25\%}$** | L/s | 0.84±0.44 | 1.02±0.52 | 0.77±0.26 | 0.1209 |
| **FEF$_{50\%}$** | L/s | 3.14±1.06 | 3.45±0.86 | 2.99±0.80 | 0.1822 |
| **FEF$_{75\%}$** | L/s | 5.03±1.19 | 5.33±1.08 | 5.09±0.84 | 0.5758 |
| **PEF** | L/s | 5.56±1.55 | 5.91±1.11 | 5.58±0.93 | 0.5121 |
| **Volumes and capacities** | | | | | |
| **SVC** | L | 2.74±0.61 | 3.24±0.41 | 3.02±0.53 | 0.0115*[**a**] |
| **TGV** | L | 2.64±0.41 | 2.83±0.46 | 2.97±0.50 | 0.1054 |
| **RV** | L | 1.75±0.29 | 1.90±0.33 | 2.04±0.30 | 0.0223*[**b**] |
| **TLC** | L | 4.47±0.59 | 5.10±0.58 | 5.05±0.68 | 0.0050*[**ab**] |
| **ERV** | L | 0.83±0.33 | 0.90±0.34 | 0.87±0.43 | 0.8214 |
| **IRV** | L | 1.37±0.43 | 1.66±0.41 | 1.54±0.40 | 0.0856 |
| **IC** | L | 1.92±0.41 | 2.34±0.39 | 2.15±0.42 | 0.0075*[**a**] |
| **Airway resistances** | | | | | |
| **sRaw** | kPa*s | 4.70±1.32 | 5.75±1.18 | 5.62±1.16 | 0.0227*[**a**] |
| **Ratios** | | | | | |
| **FEV$_1$/FVC** | absolute value | 0.78±0.04 | 0.77±0.03 | 0.77±0.03 | 0.5511 |
| **FEV$_1$/SVC** | absolute value | 0.77±0.04 | 0.76±0.05 | 0.75±0.05 | 0.4862 |
| **RV/TLC** | absolute value | 0.39±0.07 | 0.37±0.05 | 0.41±0.04 | 0.0719 |

For abbreviations, see Table 1. Quantitative data were mean±SD. Qualitative data (obesity and menopause status) were number (%).

*p (analysis of variance) < 0.05: comparison between the 3 classes. Tukey test: [**a**]$C_1$ vs. $C_2$; [**b**]$C_1$ vs. C3; [**c**]$C_2$ vs. $C_3$.

**p (Pearson Chi-square) < 0.05: comparison between the 3 classes. Mac-Nemar test: [**a**]$C_1$ vs. $C_2$; [**b**]$C_1$ vs. C3; [**c**]$C_2$ vs. $C_3$.

volume curve' data, sRaw and ratios. Compared to the $G_1$, the $G_2$ had significantly higher TGV, RV and TLC. The TLC effect size was medium (Hedges' unbiased d = +0.521).

Table 2 exposes the data of the females divided into three classes of parities ($C_1$: 1–4; $C_2$: 5–8; $C_3$: 9–14). The three classes were height-, weight-, BMI- and obesity status- matched; however, compared to $C_2$, $C_3$ was older. Compared to the $C_2$, the $C_1$ and the $C_3$ included higher percentages of menopaused females. The three classes had similar values of MMEF, FEF$_{x\%}$, PEF, TGV, ERV, IRV and ratios. Compared to the $C_1$, the $C_2$ had significantly higher FEV$_1$, FVC, SVC, TLC, IC and sRaw. Compared to the $C_1$, the $C_3$ had significantly higher TLC and RV.

**Table 3. Determination coefficient ($r^2$) between anthropometric and plethysmographic data, and parity.**

| | | Total sample (n = 66) | Groups (G) | | Classes (C) | | |
|---|---|---|---|---|---|---|---|
| | | | $G_1$ | $G_2$ | $C_1$ | $C_2$ | $C_3$ |
| **Anthropometric data** | | | | | | | |
| **Age** | years | 0.0265 | 0.0269*[b] | 0.1500*[b] | 0.6889*[a] | 0.0079 | 0.0363 |
| **Weight** | kg | 0.0447 | 0.0164 | 0.0029 | 0.0162 | 0.0323 | 0.0238 |
| **BMI** | kg/m$^2$ | 0.0524 | 0.0014 | 0.0172 | 0.1098 | 0.1160 | 0.1288 |
| **Plethysmographic data** | | | | | | | |
| **FEV$_1$** | L | 0.0085 | 0.2286*[b] | 0.2292*[b] | 0.3094*[a] | 0.0084 | 0.1292 |
| **FVC** | L | 0.0076 | 0.2699*[b] | 0.1699*[b] | 0.3130*[a] | 0.0008 | 0.1849*[b] |
| **MMEF** | L/s | 0.0032 | 0.1092 | 0.0334 | 0.3115*[a] | 0.0938 | 0.0254 |
| **FEF$_{25\%}$** | L/s | 0.0178 | 0.0423 | 0.1667 | 0.0943 | 0.0019 | 0.1915*[b] |
| **FEF$_{50\%}$** | L/s | 0.0106 | 0.0711 | 0.0895 | 0.1865 | 0.0135 | 0.0221 |
| **FEF$_{75\%}$** | L/s | 0.0001 | 0.1120 | 0.0005 | 0.3489*[a] | 0.1167 | 0.0005 |
| **PEF** | L/s | 0.0001 | 0.1504*[b] | 0.0022 | 0.4056*[a] | 0.0778 | 0.0100 |
| **SVC** | L | 0.0094 | 0.3009*[a] | 0.1630*[b] | 0.3067*[a] | 0.0045 | 0.1540 |
| **TGV** | L | 0.0958*[b] | 0.0600 | 0.0092 | 0.3008*[a] | 0.1633*[b] | 0.0092 |
| **RV** | L | 0.1094*[b] | 0.0104 | 0.0001 | 0.0142 | 0.1103 | 0.0099 |
| **TLC** | L | 0.0624*[b] | 0.2800*[b] | 0.0868 | 0.2492 | 0.0623 | 0.1235 |
| **ERV** | L | 0.0043 | 0.0300 | 0.1013 | 0.3823*[a] | 0.0857 | 0.0476 |
| **IRV** | L | 0.0003 | 0.2195*[b] | 0.0036 | 0.0692 | 0.0944 | 0.0899 |
| **IC** | L | 0.0039 | 0.2759*[b] | 0.0395 | 0.1119 | 0.0335 | 0.0726 |
| **sRaw** | kPa*s | 0.0194 | 0.0789 | 0.0071 | 0.2085 | 0.0063 | 0.0504 |
| **FEV$_1$/FVC** | absolute value | 0.0292 | 0.0011 | 0.0209 | 0.0793 | 0.0979 | 0.0407 |
| **FEV$_1$/SVC** | absolute value | 0.0183 | 0.0008 | 0.0085 | 0.2067 | 0.0407 | 0.0004 |
| **RV/TLC** | absolute value | 0.0194 | 0.1340*[b] | 0.1671*[b] | 0.2185 | 0.0363 | 0.1007 |

For abbreviations, see Table 1. G1 (n = 34): parity ≤ 6. $G_2$ (n = 32): parity > 6. $C_1$ (n = 15): 1 ≤ parity ≤ 4. $C_2$ (n = 28): 5 ≤ parity ≤ 8. $C_3$ (n = 23): 9 ≤ parity ≤ 14.
*$p < 0.05$: significant $r^2$. Correlations were: [a]Clinical significant: "$r^2$" ≥ 0.30; [b]Non clinical significant: "$r^2$" < 0.30.

Table 3 presents the "$r^2$" between parity, and anthropometric and plethysmographic data. No "clinically significant" correlation was found between anthropometric or plethysmographic data, and parity of the total sample, $G_2$, $C_2$ or $C_3$ (all "$r^2$" were < 0.30). In the $G_1$, a positive "clinically significant" correlation was found between parity and SVC. In $C_1$, a negative "clinically significant" correlation was found between parity and age, and positive "clinically significant" correlations were found between parity and FEV$_1$, FVC, MMEF, FEF$_{75\%}$, PEF, SVC, TGV and ERV.

## Discussion

The main results of the present study were that the two groups of parities had similar flow-volume curve' data values, sRaw and ratios; and that the three classes of parities had similar values of MMEF, FEF$_{x\%}$, PEF, TGV, ERV, IRV and ratios. However, compared to the $G_1$, the $G_2$ had higher TGV, RV and TLC; compared to the $C_1$, the $C_2$ had higher FEV$_1$, FVC, SVC, TLC, IC and sRaw; and compared to the $C_1$, the $C_3$ had higher TLC and RV. The clinical significance of this study is clear: high parity is associated with a tendency towards lung-hyperinflation and two females of similar age and height, but of two different parities, have different lung static volumes.

The respiratory phenomenon highlighted in this study may be an evidence of a more general aging phenomenon related to multiparity [39]. The link between parity and longevity is widely discussed in the scientific literature in terms of "selection pressure" [39]. Above all, it is the lung which both generates and undergoes the repercussions of the multiple physio-pathological episodes of the female life. Indeed, it is known that multiparity has adverse health effects, with a high risk of heart disease and renal cancer [40, 41]. Gestation is probably an event that, when repeated, may have consequences for LFD. In females, such study is needed in order to better understand some specific factors contributing in their lifelong LFD' loss. It should be noted that, while the total fertility rate in Algeria declined from 1951 to 2015 (from 7.279 to 2.839 or 3.100 children [42]), it remained still higher in some other African countries [23] (S1 File). To the best of the authors' knowledge, the impacts of parity on healthy females' LFD has been treated in only a few publications [6, 7, 11–16], largely described in the S2 File.

## Methodology discussion

Discussion related to the study design, the applied inclusion and non-inclusion criteria, the choice of the cutoff of six parities, and some LFD influencing factors is highlighted in the S3 File.

This study presents four limitations related to the non-identification of the schooling-level, socioeconomic-level, physical-activity level, and number of caesareans. The first three factors, recognized as LFD' influencing factors [6, 11, 43, 44], may differ between the two groups and the three classes of parities, and therefore can explain the tendency towards lung-hyperinflation observed in the $G_2$, $C_2$ and $C_3$. First, compared to females with a high schooling-level, those with a low level had higher parity [20] and lower LFD [6]. However, LFD differences concerned only peripheral flows such as PEF (increase by 254 mL/s) and $FEF_{25\%}$ (increase by 150 mL/s) [6]. Secondly, in high-income countries, compared to females with a high socioeconomic-level, those with a low one had higher parity [20], lower LFD [43, 44], and a tendency towards a distal OVD [44]. Moreover, in Polish females with COPD, the $FEV_1$ decline' acceleration with increased parity ($\geq$ 4) was accounted for by the low socioeconomic-level of the latter [20]. However, it seemed that North-African females' LFD weren't influenced by the socioeconomic-level [6]. Thirdly, physical-activity level was positively correlated with some LFD (eg, $FEV_1$, FVC, $FEV_1$/SVC, PEF and $FEF_{50\%}$) [6]. Yet, no correlation existed with static lung volumes (eg, TGV, RV, TLC) [6]. Moreover, in this study, it can be speculated that included females were sedentary since only females with low physical-activity levels were included. At least, it was better to report the number of caesarians for each female and to study its correlation with LFD. In fact, caesarean section induced a decrease in abdominal muscle strength [45], which can influence the needed forced expiratory maneuvers during the plethysmographic test. A specific study about the above mentioned issue will be of a great interest in the respiratory field.

## Results discussion

LFD are very interesting markers used to define the respiratory system's ageing, since their declines are a predictor of mortality [46]. But is-it possible to distinguish between "normal aging" in relation to the natural wear of the respiratory system and "pathological aging", which is characterized by increased, above-normal, deterioration of this system? The contribution of physiology is fundamental in this context [6]. The impacts of parity on the LFD of healthy females has been treated in few publications [6, 7, 11–16], especially bearing on those aged $\geq$ 40 years [6, 7, 11–13] (S2 File). Moreover, among the aforementioned studies, only one determined plethysmographic LFD [6]. Other studies evaluated the impacts of parity on

respiratory muscle strength [17, 21] and physical function [18, 30] of healthy females, and some others included females with chronic diseases [*eg*, COPD [20], protease inhibitor phenotype [19], diabetes [31] and sleep-apnea syndrome [29]]. This study showed that the $G_2$ compared to the $G_1$, and $C_2$ or $C_3$ compared to $C_1$, had a tendency towards lung-hyperinflation. The latter, a major concern in the management of some chronic respiratory diseases [36, 47], leads to an increase in the relaxation volume due to the reduction of lung elastic retraction forces [48]. Lung-hyperinflation has deleterious clinical, functional and radiological consequences that make it a major source of impaired quality of life [48, 49].

This study's main results are intermediate among those reported in literature (S2 File). On the one hand, findings related to $FEV_1$ and FVC (compared to the $C_1$, the $C_2$ had significantly higher $FEV_1$, FVC and sRaw (Table 2)) are partially similar to those obtained in some studies [11, 14, 16]. First, it seems that younger American females ($< 50$ years) with parity $\geq 1$, compared to nulliparous ones, had larger $FEV_1$ and FVC [11] (S2 File). Secondly, it appears that increased parity (primigravida, nullipara, primipara, $para_2$ and $para_3$) favorably affects the LFD of Nigerians, with increases in both $FEV_1$ and FVC [14]. Thirdly, as parity increases (primigravida, nullipara, primipara, $para_2$ and $para_3$), Nigerians' FVC and $FEV_1$ also increased across all studied positions [16]. Fourthly, a North-African study showed that high parity leads to a decrease in total airway conductance [6]. On the other hand, the present study findings related to $FEV_1$ FVC and peripheral airway flows (Table 2) are partially opposite with these of others concluding that multiparity negatively impacts LFD [6, 7, 12, 13, 15] (S2 File). First, a study including females aged $\geq 60$ years [7], concluded that compared to females with a parity $\leq 4$, those with a parity $> 4$ had lower $FEV_1$, $FEV_1/FVC$, MMEF and PEF. Secondly, another study including females aged $\geq 40$ years [6], concluded that multiparity leads to a tendency towards an OVD with a decrease in $FEV_1/FVC$. Thirdly, compared to Brazilian nulliparous females aged $< 25$ years, those with a parity $\geq 1$ had lower MMEF during the first trimester and lower PEF during the third trimester [15]. Fourthly, it appears that parity accelerated lung-aging, with an increase of one parity rising the estimated lung-age by 1.2 years [12, 13]. Finally, findings related to TGV, RV and TLC (Tables 2 and 3) are totally opposite with these of the only study that evaluated static volumes [6] (S2 File). While this study's findings suggested that multiparity is associated with a tendency towards lung-hyperinflation (Tables 2 and 3), a Tunisian study showed that the two groups ($\leq 3$; $> 4$) and the three classes (0–2; 3–4; $> 5$) of parities had similar TGV, VR and TLC data (expressed as percentage of predicted values), meaning the lack of a trend towards lung-hyperinflation [6].

The present study correlations between parity and LFD (Table 3) are intermediate between those reported in the literature [6, 7, 11] (S2 File). On the one hand, similar to this study, where no correlation was found between the parities of the total sample, $G_2$, $C_2$ and $C_3$, and LFD (Table 3), one study also reported no correlation between the parity (mean not reported) and the flow-volume curve data (*ie*, FVC, $FEV_1$, MMEF, PEF) [7]. Moreover, another study reported no correlation between parity (mean: 4±2) and static lung volumes (*ie*, TGV, TLC, RV) [6], and reported negative but not "clinically significant" correlations between parity and FVC, $FEV_1$, PEF, $FEF_{75\%}$, $FEF_{50\%}$, MMEF and SVC (all "$r^2$" were $< 0.30$) (S2 File). In addition, positive but not "clinically significant" correlations were noted between parity of 397 Caucasians (mean not reported) and $FEV_1$ (S2 File) [11]. On the other hand, similar to this study, where "clinically significant" correlations were found between parity and $G_1$' SVC and $C_1$' some LFD (Table 3), another study identified a negative "clinically significant" correlation between parity and $FEV_1/SVC$ ($r^2 = 0.5334$) [6]. Since parity wasn't correlated either to weight or to BMI (Table 3), its impact on LFD appears to be independent from those two parameters.

The discrepancy between reported results in literature may be due to some different obstetric/anthropometric characteristics of the included females (S2 File): different parity means

[4±2 [6], 5±3 [12, 13], 6±3 (this study)], different age ranges [21–28 [16], 18–92 [11], 19–90 [12, 13], 40–74 (this study), 45–90 [6], 60–96 [7]], different applied cutoff for parities [1 [11, 15], 3 [6], 4 [7], 6 (this study)], different groups of parities [n = 2 [6, 7, 11, 15], n = 3 [6], n = 4 [15], n = 5 [14, 16]].

## How to explain the impacts of parity on LFD?

During healthy pregnancy, respiratory function is affected through both biochemical and mechanical pathways [50, 51]. Throughout gravidity, spirometry remains within normal ranges (*ie*; unchanged FVC, $FEV_1$, and $FEV_1$/FVC, unchanged or a modest increase of PEF [50–52]). Conversely, lung volumes endure for most variations: ERV progressively declines during the second half of gestation since RV decreases [50–52]. TGV then diminishes while IC rises in the same degree in order to conserve stable TLC [50–52]. Bronchial resistance rises whereas respiratory conductance decreases during gestation [50–52]. Total pulmonary and airways resistances have a tendency to decline in late gestation as a result of hormonally induced relaxation of tracheobronchial tree smooth muscles [51]. What happens with increasing parity? With increasing parity, the rise in TGV, RV and TLC (Tables 1 and 2), and therefore the tendency towards lung-hyperinflation, can be interpreted as an aging index of the ventilatory mechanics, or as an indirect sign towards an OVD and/or an expiratory muscle weakness [36, 53]. The tendency towards lung-hyperinflation can be explained by at least the four following hypothesizes:

1. *Anatomical changes*: during gestation, the progressive increase of the uterus volume is the main reason for lung volume and chest wall changes (*eg*, elevation of the diaphragm, altered thoracic shape) [50–51, 54]. The diaphragm elevation induced two phenomena: *i)* earlier closure of the lower airways with consequent reduction of TGV and ERV; *ii)* shorter chest height, but increase of the other thoracic dimensions in order to maintain constant TLC [50–52]. Gestation is also accompanied by changes in the mucosa of the upper and lower airways with the appearance of inflammatory phenomena [54, 55]. Thus, the effects of these changes, can accumulate with repeated gestations. Chest circumference may increase and hypotrophy of the respiratory muscles may develop. This will explain the decline of the maximal inspiratory pressure with high parity (S2 File) [56].

2. *Hormonal changes*: during gestation, the physiological adaptation of hormonal (progesterone, estrogen and prostaglandins) profiles is the foremost cause of ventilatory changes in respiratory function [50, 51]. Progesterone modifies the airways' smooth muscle tone inducing a bronchodilator effect [50, 51]. Estrogen upsurges the number and the sensitivity of progesterone receptors within several nervous areas (*eg*, hypothalamus, medulla, and central neuronal respiratory-related areas) [50, 51]. Prostaglandin $F_{2\alpha}$ rises airway resistance by bronchial smooth muscle constriction, whereas a bronchodilator effect can be a consequence of prostaglandins $E_1$ and $E_2$ [51]. The aforementioned hormonal changes are related to LFD variations [50, 57]. With repeated gestations, it can be speculated that hormonal changes persist and accumulate. During the ageing process, aging-induced hormonal changes can modify LFD [58] (*eg*, elderly female' cortisol secretion determined the rate of the lung-aging [59]). Since gestation is experienced as a stressful situation, hormonal changes can increase in multiparous females.

3. *Biochemical changes*: the natural damage of elastin with age, contributing to the LFD' decline, is less accelerated in females with a moderate deficiency in protease inhibitor and having a high parity [60]. This has been attributed to an improvement in elastin turnover in these females with high parity [60]. This finding has not been proven in females with normal protease inhibitor phenotype and high parity [60].

4. *Bronchial hyperreactivity*: with gestation, there is a decrease in bronchodilator factors ($\beta_2$-adrenergic receptors and adenylyl-cyclase activity) in favor of an increase in bronchoconstrictor ones (prostaglandin $F_{2\alpha}$ and cyclic guanosine monophosphatectively) [55]. These effects can accumulate with repeated gestations and partially explain the tendency towards lung-hyperinflation.

In conclusion, high parity is associated with a tendency towards lung-hyperinflation. In females, parity should be considered, along with sex and anthropometric data, as a major determinant of LFD.

## Supporting information

**S1 File. Entity, Code, Year, "Estimates, 1950–2015: Demographic Indicators—Total fertility (live births per woman) (live births per woman)".**
(DOCX)

**S2 File. Studies evaluating the effects of parity on lung function data (LFD) of healthy females: Designs and results.**
(DOCX)

**S3 File. Appendix: Discussion.**
(DOCX)

**S4 File. Spirometric data of the 66 Algerian females.** Data are "Excel file".
(XLSX)

## Acknowledgments

Authors wish to thank professor Farida Hellal (Freelance Translators) for her invaluable contribution in the improvement of the quality of the writing in the present paper.

## Author Contributions

**Conceptualization:** Abdelbassat Ketfi, Merzak Gharnaout, Helmi Ben Saad.

**Data curation:** Abdelbassat Ketfi, Helmi Ben Saad.

**Formal analysis:** Abdelbassat Ketfi, Leila Triki, Merzak Gharnaout, Helmi Ben Saad.

**Investigation:** Abdelbassat Ketfi, Merzak Gharnaout, Helmi Ben Saad.

**Methodology:** Abdelbassat Ketfi, Leila Triki, Merzak Gharnaout, Helmi Ben Saad.

**Project administration:** Helmi Ben Saad.

**Software:** Helmi Ben Saad.

**Supervision:** Abdelbassat Ketfi, Merzak Gharnaout, Helmi Ben Saad.

**Validation:** Abdelbassat Ketfi, Leila Triki, Merzak Gharnaout, Helmi Ben Saad.

**Visualization:** Abdelbassat Ketfi, Helmi Ben Saad.

**Writing – original draft:** Abdelbassat Ketfi, Leila Triki, Merzak Gharnaout, Helmi Ben Saad.

**Writing – review & editing:** Abdelbassat Ketfi, Leila Triki, Merzak Gharnaout, Helmi Ben Saad.

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
