## [Decision Letter · Decision Letter 0]

21 Aug 2019

PONE-D-19-16514

The impacts of parity on lung function data (LFD) of healthy females aged 40 years and more issued from an upper middle income country (Algeria): a comparative study

PLOS ONE

Dear Pr Ben Saad,

Thank you for submitting your manuscript to PLOS ONE. After careful consideration, we feel that it has merit but does not fully meet PLOS ONE’s publication criteria as it currently stands. Therefore, we invite you to submit a revised version of the manuscript that addresses the points raised during the review process.

The review of your manuscript reveals that you have significantly digressed from the main topic or focus of your title. Both reviewers feel that not enough has been done or discussed to address the factor of 'parity' in your study although that is the main theme of your manuscript. Besides, serious concerns about the methodology and conclusions drawn have also been raised by the reviewers. We would therefore advise you to carefully go through the comments/suggestions made by both the reviewers and address them point-by-point through a revised manuscript.

We would appreciate receiving your revised manuscript by Oct 05 2019 11:59PM. To enhance the reproducibility of your results, we recommend that if applicable you deposit your laboratory protocols in protocols.io, where a protocol can be assigned its own identifier (DOI) such that it can be cited independently in the future. For instructions see: http://journals.plos.org/plosone/s/submission-guidelines#loc-laboratory-protocols

We look forward to receiving your revised manuscript.

Kind regards,

Koustubh Panda, M. Tech., Ph.D

Academic Editor

PLOS ONE

Journal Requirements:

I have read the journal's policy and the authors of this manuscript have the following competing interests: [Helmi BEN SAAD reports personal fees from AstraZeneca, Boehringer Ingelheim, INPHA-MEDIS, Teriak, Chiesi, SAIPH and HIKMA. The remaining authors declare that they have no conflicts of interest concerning this article.]

Reviewers' comments:

Reviewer's Responses to Questions

**Comments to the Author**

1. Is the manuscript technically sound, and do the data support the conclusions?

Reviewer #1: Yes

Reviewer #2: Partly

2. Has the statistical analysis been performed appropriately and rigorously? 

Reviewer #1: Yes

Reviewer #2: No

3. Have the authors made all data underlying the findings in their manuscript fully available?

Reviewer #1: Yes

Reviewer #2: Yes

4. Is the manuscript presented in an intelligible fashion and written in standard English?

Reviewer #1: Yes

Reviewer #2: Yes

5. Review Comments to the Author

Reviewer #1: Overall this is a well written paper using sound methodologies. There are a few issues, however, in its current form.

Major comments:

While the authors spend a significant amount of time attempting to explain why they chose parity less than or equal to 6 as their definition of low parity, it still seems like it includes a high number. The authors mention in their introduction that parity is high in Africa but give reference to numbers which would be otherwise considered in the “low parity group,”( i.e everyone less than 6). Also in the results section they mention previous studies and different parity cutoffs which are all lower than the current study.

- Why not divide into 3 groups? 1-4, 5-8, 9-12 for instance.

- Can there be more information on the health of the females. You mention education, access to food and education, and that the average was 0.75. Then you mention in the discussion that SL, SEL, PAL cannot be identified. Why was this? Is this information included in the study?

- Higher parity can be associated with lower economic status. How do you adjust for that in your study?

Minor comments:

- “Since the different aspects of LFD’ decline are still inescapable [5], and since, the determination of numerous factors of LFD’ decline is not yet closed” – I don’t understand what closed means

- “while the total fertility rate in Algeria declined from 1951 to 2015 (from 7.279 to 2.839 or 3.100 children [42]), it remained still higher in some author African countries” -> I think you mean “other”

Reviewer #2: RE: PONE-D-19-16514 -- Parity and ageing of the respiratory system

Studies evaluating the impacts of parity on lung function data (LFD) of healthy females present controversial conclusions. Respiratory aging, can be estimated by lung function data (LFD), recognized as predictors of mortality and morbidity. These relationships are based on the subjects’ ethnicity and are included as influencing factors [eg, gender, age, height, weight and body mass index (BMI)].

Parity has been proposed as an additional influence in some studies. Contrary to European and North American studies, parity is of particular impact in low- and lower middle-income countries such as in Africa. In several African countries, recent values of parity were higher than six. A few studies, published between 1999 and 2018, raised the impacts of parity on healthy females’ LFD [USA (n=1), Tunisia (n=3), Nigeria (n=2) and Brazil (n=1)] with conflicting results. Some studies concluded that high parity was positively associated LFD while others concluded that it was associated with negative effects on LFD.

Thus, the purpose of this investigation was to compare lung function data (LFD) of two groups of healthy females divided according to their parities, and to determine the relationship between parity and LFD.

Methods. A medical questionnaire was administered and anthropometric data were determined. Parity was introduced as numeric and as dichotomous [G1 ≤ 6 parity; G2 >6 parity]. LFD (plethysmography, specific airway resistance (sRaw)] was performed. Correlation-coefficient (r) and Student’s t-test were used, respectively, to evaluate therelationships between parity and LFD (absolute values) and to compare the two groups’ mean±SD quantitative data.

Main results: Compared to GI, G2 had a lower mean body mass index (30.vs. 27.6) and a higher parity (4 vs. 10). G1 and G2 had similar values of forced expiratory volume in 1 s (FEV1), forced- and low- vital capacity (FVC, SVC), maximal mid-expiratory flow, forced expiratory flow at x% of FVC, peak expiratory flow, expiratory and inspiratory reserve volumes, inspiratory capacity, sRaw, FEV1/FVC, FEV1/SVC, and residual volume/total lung capacity (RV/TLC). Compared to G1, G2 had significantly higher thoracic gas volume, RV and TLC. In the total sample, significant positive correlations were found between parity and TGV (r=0.31), RV (r=0.33) and TLC (r=0.25).

The authors concluded that increasing parity induces a tendency towards lung-hyperinflation.

General comments:

A key goal of this study should be to describe and explain the differences between high and low parity women, and explain the differences found between the two cohorts and the mechanisms that would explain the findings. Instead, the authors launch into a lengthy disquisition of why lung function testing is obtained in general, how lung function changes with age and other factors (smoking, environment, etc), how cohort values are compared to controls, and a detailed description of differences in lung function amongst European, American and northern (the Sahel) and sub-Saharan African ethnic groups. This is interesting material but has little to do with the main purpose of the study which was ostensibly to assess the effects of parity on lung function (and what is implied in the title of the paper). The authors should have primarily focused on this topic. The rest of their discussion is better suited for a textbook or monograph on the epidemiology of lung function.

Not only that, but the paper is needlessly prolonged by repeating in the discussion much of what’s already stated in the introduction. The explanation for the changes in LFD is only briefly explained in a paragraph ending on p. 23. Instead of placing most of this information in the supplementary material it should be included as part of the main discussion – that is the main point of the paper.

Specific comments:

The paper needs grammatical and stylistic corrections. I will list a few:

1. Pages and lines should be numbered – makes changes easily traceable.

2. The authors should reconsider the weightedness and clinical importance of their correlation coefficients. From a clinically relevant standpoint, the correlation coefficient should be squared: an r-squared value of >0.3 is considered to be clinically relevant. Thus r=0.3 becomes 0.09, which is not clinically meaningful; r=0.4 becomes 0.16, also not relevant. In other words, what may be statistically significant may not be clinically significant or relevant.

3. Intro., l. 8, “…next line: “…ongoing...” instead of “… not yet closed…”

4. Next line: Instead of “…guaranteed...”, state “…has been validated…”

5. Intro, next page, l. 5: Chad is repeated in same sentence.

6. Same page, l. 7: For Tunisia there are 4 references cited, not 3.

7. Intro, last paragraph, l. 1: “In view of…” instead of “…in front of…”

8. Under plethysmographic measurements, l. 1; should read “Lung volume data were determined by… plethysmography (Body-box 5500…)".

9. Under discussion, last para, last line: “… still higher in other African countries.”

10. Under Methodology discussion, l. 8: “…who reported being healthy…”

11. Same section, next 2 pages: Beginning with “The non-inclusion criteria were respected.” all the way through the next page, ending with “… can’t be explained by their menopause status” should be deleted, as it is repetitious from the introduction and also not relevant to the discussion itself.

12. Results discussion, last 2 pages: Beginning with “The present study correlations…”, the authors just regurgitate what is already listed in the supplementary tables – this information should be transferred to the results section and not repeated here. Rather, the authors should expand on and explain their findings from a physiological standpoint, that is, how are the findings explained. This has much to do with physiologic differences of the thoracic cage between men and women and how pregnancy affects these mechanical properties during pregnancy and with repeated pregnancies. This would add a unique aspect to their discussion because there is not much information on the effects of multiparity on respiratory mechanics.

13. Again, these r-values should be squared to determine if they are truly clinically significant – many of the LFD changes will turn out to be not significant or relevant to parity.

14. Last page, last para.: This brief paragraph should be greatly expanded to explain the effects of multiparity on respiratory function (here, not in the supplement), particularly in regards to the lung hyperinflation – how is this event linked to changes in the thoracic cage? In fact, this expanded discussion should nearly completely replace the bloated epidemiologic data from different countries listed by the authors. Pregnancy likely affects the fundamental changes that occur in the respiratory system in a common way, with only subtle differences amongst ethnic/national groups.

In short, the information provided here is interesting, and relatively new and should be reported, but in a greatly revised form. It can be presented in a cleaner, more concise manner with greater emphasis placed on the physiologic explanations rather than just reporting epidemiologic data from different countries. The latter aspect can be considerably shortened. Finally, the clinical relevance of statistically significant correlations should be re-considered in a clinically relevant manner.

6. PLOS authors have the option to publish the peer review history of their article (what does this mean?). If published, this will include your full peer review and any attached files.

Reviewer #1: No

Reviewer #2: No

<gdiv></gdiv>

---

## [Author Response · Author response to Decision Letter 0]

25 Sep 2019

TO THE ACADEMIC EDITOR

Dear EDITOR,

Thank you for giving us the chance to review and improve our paper. Please find below the responses to the two reviewers questions/suggestions. Sincerely yours.

TO REVIEWER #1

Dear Reviewer,

Thank you for your comments. Please find below the responses to your questions/suggestions. 

Sincerely,

REMARK N°1

Overall this is a well written paper using sound methodologies. There are a few issues, however, in its current form. 

CORRECTIVE ACTION

All your suggestions were applied.

REMARK N°2 a

*While the authors spend a significant amount of time attempting to explain why they chose parity less than or equal to 6 as their definition of low parity, it still seems like it includes a high number. The authors mention in their introduction that parity is high in Africa but give reference to numbers which would be otherwise considered in the “low parity group,”( i.e everyone less than 6). 

**Also in the results section they mention previous studies and different parity cutoffs which are all lower than the current study. Why not divide into 3 groups? 1-4, 5-8, 9-12 for instance.

RESPONSE

*Needed changes were applied over all the paper.

**We have takin into account your suggestion to divide the population into three 3 classes: 

Parity 1-4: n=15

Parity 5-8: n=28

Parity 9-14: n=23

CORRECTIVE ACTION

*The following sentence was deleted from the “Introduction” “[eg, 2.570, 3.229, 4.673, 4.927, 5.374 and 5.749, respectively, in Middle-, Southern-, Northern-, Eastern-, Sub-Saharan-, and Middle- Africa]”.

**In the “Methods” section, we have added that our population was divided into two groups (Table 1) and into three classes (Table 2) of parity.

*We have also applied all needed changes in the text (Methods, Results, Discussion, Tables).

REMARK N°2 b

Can there be more information on the health of the females. You mention education, access to food and education, and that the average was 0.75. Then you mention in the discussion that SL, SEL, PAL cannot be identified. Why was this? Is this information included in the study?

RESPONSE

In the manuscript (L185-187), we have noted that “The population was relatively homogeneous and considered as belonging to the middle class with an elevated human development index of 0.75 [27].” The information was retrieved from the following reference (ref 27): Human development index in Algeria. Therefore, SL, SEL, PAL weren’t objectively determined.

CORRECTIVE ACTION

Education (schooling level) was addressed as a limitation study (L303-306): “First, compared to females with a high schooling-level, those with a low level had higher parity [20] and lower LFD [6]. However, LFD differences concerned only peripheral flows such as PEF (increase by 254 mL/s) and FEF25% (increase by 150 mL/s) [6].”

REMARK N°2 c

Higher parity can be associated with lower economic status. How do you adjust for that in your study?

RESPONSE

We agree with your remark. The association between high parity and lower socioeconomic level was highlighted as a study limitation (L306-311).

CORRECTIVE ACTION

The following sentence is noted in the paper (L306-311): “Secondly, in high-income countries, compared to females with a high socioeconomic-level, those with a low one had higher parity [20], lower LFD [43, 44], and a tendency towards a distal OVD [44]. Moreover, in Polish females with COPD, the FEV1 decline’ acceleration with increased parity (≥ 4) was accounted for by the low socioeconomic-level of the latter [20]. However, it seemed that North-African females’ LFD weren’t influenced by the socioeconomic-level [6].”

REMARK N°3 a

Since the different aspects of LFD’ decline are still inescapable [5], and since, the determination of numerous factors of LFD’ decline is not yet closed” – I don’t understand what closed means

RESPONSE

The sentence was corrected (as suggested by the 2nd reviewer: please see his remark 9c)

CORRECTIVE ACTION

“ongoing” replaced “not yet closed” L110.

REMARK N°3 b

while the total fertility rate in Algeria declined from 1951 to 2015 (from 7.279 to 2.839 or 3.100 children [42]), it remained still higher in some author African countries” -> I think you mean “other”

RESPONSE

Correction done.

CORRECTIVE ACTION

Please see L292

TO REVIEWER #2

Dear Reviewer,

Thank you for your comments. Please find below the responses to your questions/suggestions. Sincerely yours.

REMARK N°1

Studies evaluating the impacts of parity on lung function data (LFD) of healthy females present controversial conclusions. Respiratory aging, can be estimated by lung function data (LFD), recognized as predictors of mortality and morbidity. These relationships are based on the subjects’ ethnicity and are included as influencing factors [eg, gender, age, height, weight and body mass index (BMI)].

Parity has been proposed as an additional influence in some studies. Contrary to European and North American studies, parity is of particular impact in low- and lower middle-income countries such as in Africa. In several African countries, recent values of parity were higher than six. A few studies, published between 1999 and 2018, raised the impacts of parity on healthy females’ LFD [USA (n=1), Tunisia (n=3), Nigeria (n=2) and Brazil (n=1)] with conflicting results. Some studies concluded that high parity was positively associated LFD while others concluded that it was associated with negative effects on LFD.

Thus, the purpose of this investigation was to compare lung function data (LFD) of two groups of healthy females divided according to their parities, and to determine the relationship between parity and LFD.

Methods. A medical questionnaire was administered and anthropometric data were determined. Parity was introduced as numeric and as dichotomous [G1 ≤ 6 parity; G2 >6 parity]. LFD (plethysmography, specific airway resistance (sRaw)] was performed. Correlation-coefficient (r) and Student’s t-test were used, respectively, to evaluate the relationships between parity and LFD (absolute values) and to compare the two groups’ mean±SD quantitative data.

Main results: Compared to GI, G2 had a lower mean body mass index (30.vs. 27.6) and a higher parity (4 vs. 10). G1 and G2 had similar values of forced expiratory volume in 1 s (FEV1), forced- and low- vital capacity (FVC, SVC), maximal mid-expiratory flow, forced expiratory flow at x% of FVC, peak expiratory flow, expiratory and inspiratory reserve volumes, inspiratory capacity, sRaw, FEV1/FVC, FEV1/SVC, and residual volume/total lung capacity (RV/TLC). Compared to G1, G2 had significantly higher thoracic gas volume, RV and TLC. In the total sample, significant positive correlations were found between parity and TGV (r=0.31), RV (r=0.33) and TLC (r=0.25).

The authors concluded that increasing parity induces a tendency towards lung-hyperinflation.

RESPONSE

No corrective action is needed in this stage. Please note that we have applied all your asked corrections.

REMARK N°2a

A key goal of this study should be to describe and explain the differences between high and low parity women, and explain the differences found between the two cohorts and the mechanisms that would explain the findings. 

Instead, the authors launch into a lengthy disquisition of why lung function testing is obtained in general, how lung function changes with age and other factors (smoking, environment, etc), how cohort values are compared to controls, and a detailed description of differences in lung function amongst European, American and northern (the Sahel) and sub-Saharan African ethnic groups. 

This is interesting material but has little to do with the main purpose of the study which was ostensibly to assess the effects of parity on lung function (and what is implied in the title of the paper). 

The authors should have primarily focused on this topic. The rest of their discussion is better suited for a textbook or monograph on the epidemiology of lung function.

RESPONSE

Several changes were performed in the Discussion section. 

CORRECTIVE ACTION

*The paper is now primarily focused on the effects of parity on lung function data. 

**Several parts from the “Methodology discussion” subsection was moved to the S3 File.

REMARK N°2b

Not only that, but the paper is needlessly prolonged by repeating in the discussion much of what’s already stated in the introduction. 

The explanation for the changes in LFD is only briefly explained in a paragraph ending on p. 23. Instead of placing most of this information in the supplementary material it should be included as part of the main discussion – that is the main point of the paper.

RESPONSE

Several changes were performed in the Discussion.

CORRECTIVE ACTION

The answer to the following question “How to explain the impacts of parity on LFD?” is moved from the S3 File to the main manuscript (L381-428).

REMARK N°3a

Pages and lines should be numbered – makes changes easily traceable.

CORRECTIVE ACTION

Pages and lines are added.

REMARK N°3b

The authors should reconsider the weightiness and clinical importance of their correlation coefficients. From a clinically relevant standpoint, the correlation coefficient should be squared: an r-squared value of >0.3 is considered to be clinically relevant. Thus r=0.3 becomes 0.09, which is not clinically meaningful; r=0.4 becomes 0.16, also not relevant. In other words, what may be statistically significant may not be clinically significant or relevant.

RESPONSE

All your suggestions were applied.

CORRECTIVE ACTION

*In the ‘statistical analysis” subsection, the following sentence was added (L239-241): “Determination-coefficient (r2 = square of the Pearson product-moment correlation-coefficient) evaluated the associations between parity, and plethysmographic and anthropometric data. “r2” was considered as “clinically significant” when it was > 0.30 [37]” 

**Table 3 exposes the “Determination coefficient (r2) between anthropometric and plethysmographic data, and parity.” Only “clinically significant” correlations were considered.

***All required changes were applied in the main manuscript.

REMARK N°3c

Intro., l. 8, “…next line: “…ongoing...” instead of “… not yet closed…”

RESPONSE

Agree with you

CORRECTIVE ACTION

Done L110.

REMARK N°3d

Next line: Instead of “…guaranteed...”, state “…has been validated…”

RESPONSE

Agree with you

CORRECTIVE ACTION

Done L112.

REMARK N°3e

Intro, next page, l. 5: Chad is repeated in same sentence.

RESPONSE

Agree with you

CORRECTIVE ACTION

The second ‘Chad” was deleted (L129).

REMARK N°3f

Same page, l. 7: For Tunisia there are 4 references cited, not 3.

RESPONSE

References 12 (Spirometric “Lung Age” estimation for North African population. Egyptian Journal of Chest Diseases and Tuberculosis. 2014;63(2):491-503) and 13 (Estimated lung age in healthy North African adults cannot be predicted using reference equations derived from other populations. Egyptian Journal of Chest Diseases and Tuberculosis. 2013;62(4):789-804) are derived from the same project and described the same population.

CORRECTIVE ACTION

Only reference 12 was kept.

REMARK N°3g

Intro, last paragraph, l. 1: “In view of…” instead of “…in front of…”

RESPONSE

Agree with your suggestion.

CORRECTIVE ACTION

Done L143.

REMARK N°3h

Under plethysmographic measurements, l. 1; should read “Lung volume data were determined by… plethysmography (Body-box 5500…)".

RESPONSE

Agree with your suggestion.

CORRECTIVE ACTION

The following sentence was added (L218-219): “LFD were determined by one qualified person (AK in the authors’ list) via a plethysmograph (Body-box 5500, MediSoft, Belgium). The latter was calibrated each morning.”

REMARK N°3i

Under discussion, last para, last line: “… still higher in other African countries.”

RESPONSE

Agree with your suggestion.

CORRECTIVE ACTION

Done L292.

REMARK N°3k

Under Methodology discussion, l. 8: “…who reported being healthy…”

RESPONSE

Agree with your suggestion.

CORRECTIVE ACTION

Note: the “Methodology discussion“ subsection was moved to the S3 File.

REMARK N°3l

Same section, next 2 pages: Beginning with “The non-inclusion criteria were respected.” all the way through the next page, ending with “… can’t be explained by their menopause status” should be deleted, as it is repetitious from the introduction and also not relevant to the discussion itself.

RESPONSE

We have considered your suggestion and we agree with it.

CORRECTIVE ACTION

In order to shorten the manuscript, all the long sentence “The non-inclusion criteria were respected……… can’t be explained by their menopause status” was moved to the S3 File.

REMARK N°3m

*Results discussion, last 2 pages: Beginning with “The present study correlations…”, the authors just regurgitate what is already listed in the supplementary tables – this information should be transferred to the results section and not repeated here. 

**Rather, the authors should expand on and explain their findings from a physiological standpoint, that is, how are the findings explained. This has much to do with physiologic differences of the thoracic cage between men and women and how pregnancy affects these mechanical properties during pregnancy and with repeated pregnancies. 

***This would add a unique aspect to their discussion because there is not much information on the effects of multiparity on respiratory mechanics.

RESPONSE

We have taken into account all your suggestions.

CORRECTIVE ACTION

.In order to improve the scientific quality of our paper, we have added a new table (S2 File) which exposes the main results of the seven studies including healthy females and evaluating the effects of parity on lung function data. We hope that this table will facilitate the interpretation of our data taking into account those of the literature.

*We have avoided redundancy between text and tables.

**In the revised version, we tried to explain our findings from a physiological standpoint. For that reason, we have added the following sentence (L381-428) related to how pregnancy affects the mechanical properties during pregnancy and with repeated pregnancies. 

“How to explain the impacts of parity on LFD? 

During healthy pregnancy, respiratory function is affected through both biochemical and mechanical pathways [50, 51]. Throughout gravidity, spirometry remains within normal ranges (ie; unchanged FVC, FEV1, and FEV1/FVC, unchanged or a modest increase of PEF [50-52]). Conversely, lung volumes endure for most variations: ERV progressively declines during the second half of gestation since RV decreases [50-52]. TGV then diminishes while IC rises in the same degree in order to conserve stable TLC [50-52]. Bronchial resistance rises whereas respiratory conductance decreases during gestation [50-52]. Total pulmonary and airways resistances have a tendency to decline in late gestation as a result of hormonally induced relaxation of tracheobronchial tree smooth muscles [51]. What happens with increasing parity? With increasing parity, the rise in TGV, RV and TLC (Tables 1 and 2), and therefore the tendency towards lung-hyperinflation, can be interpreted as an aging index of the ventilatory mechanics, or as an indirect sign towards an OVD and/or an expiratory muscle weakness [36, 53]. The tendency towards lung-hyperinflation can be explained by at least the four following hypothesizes:

1) Anatomical changes: during gestation, the progressive increase of the uterus volume is the main reason for lung volume and chest wall changes (eg, elevation of the diaphragm, altered thoracic shape) [50-51, 54]. The diaphragm elevation induced two phenomena: i) earlier closure of the lower airways with consequent reduction of TGV and ERV; ii) shorter chest height, but increase of the other thoracic dimensions in order to maintain constant TLC [50-52]. Gestation is also accompanied by changes in the mucosa of the upper and lower airways with the appearance of inflammatory phenomena [54, 55]. Thus, the effects of these changes, can accumulate with repeated gestations. Chest circumference may increase and hypotrophy of the respiratory muscles may develop. This will explain the decline of the maximal inspiratory pressure with high parity (S2 File) [56]. 

2) Hormonal changes: during gestation, the physiological adaptation of hormonal (progesterone, estrogen and prostaglandins) profiles is the foremost cause of ventilatory changes in respiratory function [50, 51]. Progesterone modifies the airways’ smooth muscle tone inducing a bronchodilator effect [50, 51]. Estrogen upsurges the number and the sensitivity of progesterone receptors within several nervous areas (eg, hypothalamus, medulla, and central neuronal respiratory-related areas) [50, 51]. Prostaglandin F2α rises airway resistance by bronchial smooth muscle constriction, whereas a bronchodilator effect can be a consequence of prostaglandins E1 and E2 [51]. The aforementioned hormonal changes are related to LFD variations [50, 57]. With repeated gestations, it can be speculated that hormonal changes persist and accumulate. During the ageing process, aging-induced hormonal changes can modify LFD [58] (eg, elderly female’ cortisol secretion determined the rate of the lung-aging [59]). Since gestation is experienced as a stressful situation, hormonal changes can increase in multiparous females. 

3) Biochemical changes: the natural damage of elastin with age, contributing to the LFD’ decline, is less accelerated in females with a moderate deficiency in protease inhibitor and having a high parity [60]. This has been attributed to an improvement in elastin turnover in these females with high parity [60]. This finding has not been proven in females with normal protease inhibitor phenotype and high parity [60]. 

4) Bronchial hyperreactivity: with gestation, there is a decrease in bronchodilator factors (β2-adrenergic receptors and adenylyl-cyclase activity) in favor of an increase in bronchoconstrictor ones (prostaglandin F2α and cyclic guanosine monophosphatectively) [55]. These effects can accumulate with repeated gestations and partially explain the tendency towards lung-hyperinflation.”

***We hope that the actual “Discussion” section is acceptable.

REMARK N°3n

Again, these r-values should be squared to determine if they are truly clinically significant – many of the LFD changes will turn out to be not significant or relevant to parity.

RESPONSE

Agree with your suggestion.

CORRECTIVE ACTION

As previously highlighted, all needed changes were applied (please see our answer to your Remark 9b).

REMARK N°3o

Last page, last para.: This brief paragraph should be greatly expanded to explain the effects of multiparity on respiratory function (here, not in the supplement), particularly in regards to the lung hyperinflation – how is this event linked to changes in the thoracic cage? In fact, this expanded discussion should nearly completely replace the bloated epidemiologic data from different countries listed by the authors. Pregnancy likely affects the fundamental changes that occur in the respiratory system in a common way, with only subtle differences amongst ethnic/national groups.

RESPONSE

Totally agree with your suggestion.

CORRECTIVE ACTION

*The old short paragraph was greatly expanded to explain the effects of multiparity on respiratory function.

**The paragraph aiming to answer the following question “How to explain the impacts of parity on LFD?” was moved to the main manuscript (L381-428).

REMARK N°3p

In short, the information provided here is interesting, and relatively new and should be reported, but in a greatly revised form. It can be presented in a cleaner, more concise manner with greater emphasis placed on the physiologic explanations rather than just reporting epidemiologic data from different countries. The latter aspect can be considerably shortened. Finally, the clinical relevance of statistically significant correlations should be re-considered in a clinically relevant manner.

RESPONSE

All your suggestions were accepted and applied.

CORRECTIVE ACTION

*The paper was deeply revised.

**We tried to shorten our paper.

***Physiologic explanations are advanced in the main paper.

****The clinical relevance of statistically significant correlations should were re-considered in a clinically relevant manner.

---

## [Decision Letter · Decision Letter 1]

29 Oct 2019

The impacts of parity on lung function data (LFD) of healthy females aged 40 years and more issued from an upper middle income country (Algeria): a comparative study

PONE-D-19-16514R1

Dear Dr. Ben Saad,

We are pleased to inform you that your manuscript has been judged scientifically suitable for publication and will be formally accepted for publication once it complies with all outstanding technical requirements.

With kind regards,

Koustubh Panda, M. Tech., Ph.D

Academic Editor

PLOS ONE

Additional Editor Comments (optional):

Reviewers' comments:

Reviewer's Responses to Questions

**Comments to the Author**

1. If the authors have adequately addressed your comments raised in a previous round of review and you feel that this manuscript is now acceptable for publication, you may indicate that here to bypass the “Comments to the Author” section, enter your conflict of interest statement in the “Confidential to Editor” section, and submit your "Accept" recommendation.

Reviewer #1: All comments have been addressed

Reviewer #2: All comments have been addressed

2. Is the manuscript technically sound, and do the data support the conclusions?

Reviewer #1: Yes

Reviewer #2: Yes

3. Has the statistical analysis been performed appropriately and rigorously? 

Reviewer #1: Yes

Reviewer #2: Yes

4. Have the authors made all data underlying the findings in their manuscript fully available?

Reviewer #1: Yes

Reviewer #2: Yes

5. Is the manuscript presented in an intelligible fashion and written in standard English?

Reviewer #1: Yes

Reviewer #2: Yes

6. Review Comments to the Author

Reviewer #1: The authors have made the recommended changes and the manuscript is substantially improved compared to the prior version.

Reviewer #2: Authors have addressed the main issues. Much improved manuscript.

Much of the material in the appendix has been appropriately transferred to the main text.

7. PLOS authors have the option to publish the peer review history of their article (what does this mean?). If published, this will include your full peer review and any attached files.

Reviewer #1: No

Reviewer #2: No

---

## [Editor Report · Acceptance letter]

1 Nov 2019

PONE-D-19-16514R1 

The impacts of parity on lung function data (LFD) of healthy females aged 40 years and more issued from an upper middle income country (Algeria): a comparative study 

Dear Dr. Ben Saad:

I am pleased to inform you that your manuscript has been deemed suitable for publication in PLOS ONE. Congratulations! Your manuscript is now with our production department. 

With kind regards,

on behalf of

Professor Koustubh Panda 

Academic Editor

PLOS ONE